# Effect of Increasing Supplementation Levels of Coffee Pulp on Milk Yield and Food Intake in Dual-Purpose Cows: An Alternative Feed Byproduct for Smallholder Dairy Systems of Tropical Climate Regions

Julieta Gertrudis Estrada-Flores [1], Paulina Elizabeth Pedraza-Beltrán [2], Gilberto Yong-Ángel [3], Francisca Avilés-Nova [4], Adolfo-Armando Rayas-Amor [5], Alejandra Donají Solís-Méndez [2], Manuel González-Ronquillo [2], María Fernanda Vázquez-Carrillo [2] and Octavio Alonso Castelán-Ortega [2,*]

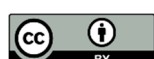

[1] Instituto de Ciencias Agropecuarias y Rurales, Universidad Autónoma del Estado de México, Instituto Literario No. 100, Toluca CP. 50000, Estado de México, Mexico; jgestradaf@uaemex.mx
[2] Faculty of Veterinary Medicine and Animal Science, Universidad Autónoma del Estado de México, Instituto Literario No. 100, Toluca CP. 50000, Estado de México, Mexico; pb_eli@yahoo.com.mx (P.E.P.-B.); adsolism@uaemex.mx (A.D.S.-M.); mrg@uaemex.mx (M.G.-R.); mvz.mafervazquez@gmail.com (M.F.V.-C.)
[3] Faculty of Veterinary Medicine and Animal Science, Universidad Autónoma de Chiapas, Carretera Emiliano Zapata Km. 8, CP. 29060, Tuxtla Gutiérrez, Chiapas, Mexico; Gilberto.yong@unach.mx
[4] Centro Universitario Temascaltepec, Universidad Autónoma del Estado de México, Temascaltepec 51300, Mexico; favilesn@uaemex.mx
[5] Food Science Deparment,Universidad Autónoma Metropolitana Campus Lerma, Av. de las Garzas No. 10, Col. El Panteón, Municipio Lerma de Villada, Estado de México, C.P. 52005, Mexico; a.rayas@correo.ler.uam.mx
* Correspondence: oacastelano@uaemex.mx; Tel.: +52-7221274130

**Abstract:** Coffee is one of the main traded commodities worldwide, unfortunately, it generates massive amounts of by-products like coffee pulp (CoP), which could be utilized as an alternative feedstuff for cattle contributing to mitigate coffee production environmental damage. The objective of this work was to evaluate the effect of increasing levels of CoP supplementation on milk production, milk composition, and grass dry matter intake (GDMI) by dual-purpose tropical cows. A 4 × 4 Latin square experimental design was conducted, where four multiparous dual-purpose Holstein x Cebu cows with an average live weight of 477 ± 7 kg and milk yield of 12.1 ± 2.7 kg/d were used. The cows grazed 10 h/d on a *Cynodon plectostachius* sward with a stocking rate of three cows/ha. All cows received 6 kg/d DM of an experimental concentrate (EC), and the treatments consisted of four supplementation levels of CoP: T1 = 0, T2 = 0.6, T3 = 0.9, and T4 = 1.2 kg DM/d, which was provided on top of the concentrate and mixed with the EC. Grass intake was determined by the n-alkanes technique. A significant difference was observed for the average total daily DM intake ($p < 0.02$). No significant differences ($p > 0.05$) were observed for milk yield, milk composition, body weight, and GDMI for all the inclusion levels of CoP. It was concluded that CoP can be included at levels of 0.6 to 0.9 kg DM/d in the diets of cows without compromising milk yield or GDMI.

**Keywords:** coffee pulp; *Cynodon plectostachius*; milk composition; tannins; polyphenols; by-pass protein; sustainable cattle production; local feed resources; coffee waste products; substitution rate

## 1. Introduction

Coffee (*Coffea arabica*) is one of the largest traded commodities in the world, ranked second, just after crude oil, among all commodities [1]. For example, in 2018, coffee world production was 9.50 million tons [2]. Approximately sixty tropical and subtropical countries produce coffee extensively, being for some of them the main agricultural export product [3]. Mexico is the seventh coffee producer in the world with more than 700,000 ha

cultivated by 280,000 farmers [4], most of which are mixed smallholder–crop–livestock farmers. It is estimated that 25 million smallholders worldwide depend on coffee production as their main source of income, that the incidence of poverty among them is high, and that the coffee trade has been identified as a major cause of biodiversity threats in tropical countries [5]. It also frequently overlooks the fact that coffee production generates massive amounts of waste and by-products, which do not undergo environmentally safe disposal. Coffee pulp (CoP) is the first by-product obtained during processing and represents 29 to 50% dry weight of the whole berry. In coffee-producing countries, coffee waste and by-products constitute a source of severe contamination and pose environmental problems because of the unsafe disposal of massive amounts of CoP, husk, and effluents leading to the pollution of water, rivers, lakes, and land around the processing units [6,7]. For example, it is estimated that the world production of fresh CoP was approximately 4.7 million tons in 2017 because for every two tons of coffee produced, one ton of CoP is obtained [8]. Therefore, there is a need to balance coffee production with the proper management and utilization of coffee by-products because a sustainable coffee industry is still a challenge. Industrial processing of coffee cherries is done to separate coffee beans by removing the shell and mucilaginous part from the cherries. During this process, the CoP is obtained, representing 29% of the dry weight of the coffee cherry [8]. The CoP could be used as a supplement to feed cattle, contributing in this way to reducing the pollution generated by coffee processing. However, large-scale utilization and management of CoP as livestock feedstuff around the world remain a challenge due to its moderate content of caffeine, free phenols, and tannins, which are known to be toxic in large quantities to many life processes [9] including the rumen microbiota [10]. However, we suggest that low to medium supplementation levels of CoP can be safely used without negative effects on animal performance. Coffee pulp is rich in soluble carbohydrates (35%), fiber (30.8%), and minerals (10.7%), but low in protein (5.2%) [3], and it also contains moderate amounts of tannins (1.8–8.5%), polyphenols (0.8–1.2%), and caffeine (1.3%) [11].

Recently, the use of some coffee by-products like spent coffee ground has been evaluated as a supplement for Holstein dairy cows with no negative effects on milk yield or fat content, and only a negligible decrease in crude protein content in milk (by 1.8%) when supplemented at a dose of 5% of the daily dry matter intake [1]. Early studies where CoP was incorporated into diets for cattle had variable success [12,13]. Limitations for the use of CoP in cattle feeding are connected to its moderate content of tannins and caffeine [14], which can affect animals when fed at high rates. Tannins are generally considered to be anti-nutritional compounds and restrict CoP from being used at more than 10% level in animal feeds [6]. Tannins can affect the acceptance of food and the utilization of nutrients by the animal; Nurfeta [12] mentioned that tannins and caffeine in the feedstuff and forage of cattle could decrease the acceptability and palatability of CoP of this by-product, while Negesse et al. [15] and Nauman et al. [16] indicated that phenolic compounds in the CoP can form insoluble structures with proteins and depress the digestibility of the organic matter (OM) and the crude protein (CP) in the cattle's diet. In contrast, CoP contains proteins, carbohydrates, and minerals that can favor its utilization in cattle feeding if used at low supplementation rates or if treated to reduce the concentration of polyphenols [11]. Moreover, Souza et al. [13] theorized that due to its conservative concentration of tannins and polyphenols, CoP can be used in the ruminants' diets because tannins can positively modulate ruminal fermentation and preserve the feed's protein from degradation by the ruminal microbiota, thereby improving animal production and serve as a sustainable way of CoP disposal. More recent evidence suggests that CoP could be an important source of valuable metabolites such as phenolic compounds with antioxidant properties and hence could be considered as a new potential functional ingredient [6], provided that these compounds can be secreted in the milk of cows fed CoP. Some early evidence in this direction was provided by Dowden [17], who identified low concentrations of theobromine in the milk of cows fed with cacao shells. Recently, Trana et al. [18] and Bonanno et al. [19] demonstrated that phenols from the diet can be transported to serum and milk confirming

Dowden's [17] findings. The use of CoP could also serve to reduce cattle feeding costs because these represent a large part of all production costs in most cattle farms. According to Alqaisi et al. [20], feed is the costliest part of dairy cattle production, normally representing 50–70% of all milk production costs [20]. Similarly, López et al. [21] reported that cattle feeding costs for smallholder dairy farmers in central Mexico ranged from 83% to 93% of total production costs. Therefore, the use of CoP in these systems may contribute to reducing feeding costs and the purchase of external inputs, revalorize a by-product currently considered as waste, and help to reduce the pollution originating from coffee production and processing.

Despite the potential benefits of the use of CoP as cattle feed, few studies have been conducted to look at the effect of this by-product on milk yield, milk composition, grass intake, and total dry matter intake in dual-purpose cows grazing tropical grasses. Therefore, the objective of the present study was to evaluate the effect of increasing supplementation levels of coffee pulp on milk yield, milk composition, and grass dry matter intake by dual-purpose cows grazing on a tropical grass species to find an appropriate supplementation rate.

## 2. Materials and Methods

The present study was carried out in the municipality of Tejupilco in the southern tropical region of central Mexico (18°45′30″N and 99°59′ 07″W), with an altitude of 1327 m. The climate is tropical subhumid, the average temperature is 24 °C, and rainfall of 1014 mm, where the rainy season is from June to November. Smallholder dual-purpose cattle production systems predominate in the region. These systems are characterized by small herds of dual-purpose cattle of up to 20 adult cows, where crosses between Holstein or Brown Swiss cattle with Cebu type of cattle predominate. The average milk yield is 10.5 kg/d/cow, and most of it is supported by concentrate supplementation because every cow is fed an average of 7.2 kg DM/d of commercial concentrate. Concentrate purchase represents up to 60% of production costs and represents the most important constraint to milk production in the system. The size of the farms ranges from 7 to 27 ha, most of which is planted with African star grass that is grazed by cattle during 8–10 h/d [22]. Coffee production is also an important activity in the Tejupilco municipality and its neighboring municipalities, Amatepec and Temascaltepec, where 475 ha of coffee (*Coffea arabica* L.) are cultivated annually [23].

### 2.1. Experimental Procedures

All experimental procedures were approved by the Committee on Bioethics and Animal Welfare of the Universidad Autónoma del Estado de México (UAEM), and no suffering was inflicted on the experimental animals. All the cows were dosed for gastrointestinal parasites before the experiment and were free of any health problems. Sun-dried CoP necessary for the experiment was obtained locally. The experiment was conducted in a smallholder experimental dairy farm from April to July 2014. A 4 × 4 Latin square experimental design was used in the present work. Four adult crossed Holstein-Cebu cows of similar milk yield and live weight were used. All cows had 100 ± 10 d of lactation. Their mean body weight at the start of the experiment was 477 ± 7 kg, and their average milk yield was 12.1 ± 2.7 kg/d. The experiment had a duration of 104 days, divided into four experimental periods of 26 days each (P1, P2, P3, and P4). The first 20 days of each period were used for diet adaptation and to wash out any carry-over effects from the previous treatment. The last six days were used for measuring milk yield, milk composition, and collecting samples of feces and feeds. The cows were milked twice daily and the milk weighed with a portable weight scale. The body weight was measured at the beginning and the end of each experimental period with a portable load-bars weight (Gallagher weighing system model W310, New Zealand). The body condition score (BCS) was recorded every week as in DEFRA [24]. Additionally, samples of milk were taken from each

cow during the measuring week and tested in the laboratory to determine their fat, protein, and total solids content with an ultrasonic milk analyzer (EKOMILK, EON Trading LLC, Stara Zagora, Bulgaria). The cows remained for 10 h/d in a sward dominated by African Star grass (*Cynodon plectostachius*), and the stocking rate was three cows per hectare. An average of ten hours is the time farmers allocate for grazing their animals [22], insecurity problems prevent them from allowing larger periods of grazing or to remain 24 h in the swards. Every cow received six kilograms per day of an experimental concentrate composed of 81% grounded maize grains (*Zea mays* L.), 10% canola cake (*Brassica napus* L.), 7% molasses, and 2% urea.

### 2.2. Treatments

Four increasing supplementation levels of sun-dried CoP were tested, TI = 0, T2 = 0.6, T3 = 0.9, and T4 = 1.2 kg CoP DM/animal/d. The CoP was fed on top of the basal diet. To facilitate the intake of the CoP by the animals, it was added and blended with the six kilograms of concentrate just before feeding and administered during the afternoon milking to prevent refusal by the animals. Each cow received each CoP level, once in each of four periods.

### 2.3. Sward Measurements

To evaluate the effect of CoP on grass dry matter intake (GDMI), the net forage accumulation (NFA) and sward height were measured during the four experimental periods. The method described by Avilés-Nova et al. [25] was used to estimate NFA, briefly: four exclusion cages of 0.72 m$^2$ × 1.0 m height per hectare were placed at random on the star grass sward and five grass height measurements were randomly carried out next to the cage, making sure the characteristics of the pasture to be measured were similar to those of the area excluded by the cage. The forage next to the cage was cut using a 0.25 m$^2$ quadrant using sheep shears. On day 26 the cage was removed and five height measurements were carried out inside the exclusion cage area and the forge was cut down to ground level using the same quadrant. NFA was calculated using Equation (1).

$$\text{NFA (kg DM/ha)} = [(\text{Initial average weight of available dry matter inside the cage on day 0} - \text{Final average weight of dry matter inside the cage on day 28}) * 40{,}000] \tag{1}$$

### 2.4. Grass Dry Matter Intake Estimation

The n-alkanes extraction technique proposed by Dove and Mayes [26] was used to measure GDMI by the experimental cows. Starting on day 10 of each experimental period, every cow received 500 mg/day of N-alkane C32, which was impregnated in 50 g of the experimental concentrate and dosed in one daily dose mixed with the experimental concentrate during the afternoon milking as in Estrada et al. [27]. Extraction of alkanes was conducted as in Mayes et al. [28] and grass intake was calculated using Equation (2).

$$\text{GDMI} = Dj \times \left(\frac{Fj}{Fi}\right) / \left(Hi - \left(\frac{Fj}{Fi}\right) * Hj\right) \tag{2}$$

where GDMI = grass dry matter intake (kg DM/d); $Dj$ = daily dose of n-alkane administered C32 (mg/d); $Fi{:}Hi$ = concentrations of odd-chain n-alkanes in forages and feces (mg/kg of DM); $Fj{:}Hj$ = concentrations of even-chain n-alkanes in feces and forages (mg/kg of DM).

### 2.5. Chemical Analysis of African Star Grass, Concentrate, and Coffee Pulp

Samples of grass for chemical analyses were obtained by hand plucking for every experimental period, and the samples were collected close to the areas grazed by the cows and at similar bite height. The samples were dried in an air forced oven at 60 °C until constant weight. Grass and experimental concentrate samples were then ground up and

analyzed for dry matter, ashes, and organic matter content. Nitrogen content was obtained using the Kjeldahl method, then crude protein (CP) expressed as nitrogen × 6.25 AOAC [29], ID 984.13. Neutral detergent fiber (NDF) and acid detergent fiber (ADF) were determined using a fiber analyzer (ANKOM200 Technology Corporation, Fairport, NY, USA) without the use of alpha-amylase. Total phenol and total tannin content in the CoP were determined by the Folin–Ciocalteu method as in Makkar [30]; tannic acid was used as a standard to create the calibration curve, so results are expressed as tannic acid equivalent. The chemical composition of the experimental concentrate, grass, and CoP is presented in Table 1.

*2.6. Analyses of Results*

Results for milk yield, milk composition, body weight, body condition score, and GDMI were analyzed by analysis of variance for a 4 × 4 Latin square experimental design as described in the lineal model below:

$$Y_{ij(k)} = \mu + C_i + P_j + T_{(k)} + \varepsilon_{ij(k)}$$

where $\mu$ is the general mean; $C_i$ is the fixed effects of cow; $P_j$ is fixed effect of the experimental period; $T_k$ is the effect of treatment; $\varepsilon_{ij(k)}$ is the residual error. The Tukey's test was applied when differences between treatment means were observed. The general linear model command of Minitab v14 [31] was used.

### 3. Results

*3.1. Chemical Composition of the Experimental Concentrate, African Star Grass, and CoP*

Table 1 depicts the chemical composition of the experimental concentrate, CoP, and African Star grass. It was observed that the CP and neutral detergent fiber contents of concentrate and CoP were within the expected range for these types of ingredients. The content of tannins in the CoP was within the low to moderate range observed in this agricultural product, and the caffeine content was low. Table 1 also shows that the CP content of the African Star grass increased from 82 g/kg DM in P1 to 155 g/kg DM in P4. In contrast, the NDF and ADF contents decreased from P1 to P4 as they passed from 721 to 598 g/kg DM and 357 to 285 g/kg DM, respectively.

**Table 1.** Chemical composition of experimental concentrate, coffee pulp, and African Star grass fodder as well as the total phenols, total tannins, and caffeine content of the coffee pulp (g/kg DM).

| Variable | DM g/kg DM | NDF g/kg DM | ADF g/kg DM | CP g/kg DM | Total Phenols g/kg DM | Total Tannins g/kg DM | Caffeine g/kg DM |
|---|---|---|---|---|---|---|---|
| Concentrate | 890 | 141 | 52 | 296 | - | - | |
| Coffee pulp | 860 | 353 | 279 | 85 | 35 | 22 | 0.1 |
| African Star grass Experimental Period | | | | | | | |
| Period 1 | 50 | 721 | 357 | 82 | | | |
| Period 2 | 55 | 657 | 334 | 97 | | | |
| Period 3 | 64 | 657 | 318 | 122 | | | |
| Period 4 | 72 | 598 | 285 | 155 | | | |
| Mean | 60.3 | 658 | 323 | 114 | | | |
| SEM | 4.8 | 21.1 | 15.1 | 16 | | | |

DM = dry matter, NDF = neutral detergent fiber, ADF = acid detergent fiber, CP = crude protein, SEM = standard error of the mean.

*3.2. Animal Response Variables*

Table 2 shows the effect of increasing the supplementation levels of CoP on milk yield, milk composition, live weight, body condition score, and grass dry matter intake (GDMI). It can be seen that no significant differences ($p > 0.05$) between treatments were

observed for any of these variables at all supplementation levels of CoP tested. In contrast, a significant difference ($p < 0.02$) for total daily DMI was observed, where the highest intake was observed for T3 and T4 in comparison with T1 and T2. The cows in T3 and T4 ate all the experimental concentrate plus the CoP, since no refusals of CoP or concentrate were recorded in any of the treatments. The results in Table 2 also show that GDMI was modest in all treatments, less than 3.6 kg DM/d. A numerical difference of more than two kilograms of milk was observed between treatments T2 and T3 (19.5% and 17.8% more milk, respectively) in comparison with T1 (control) and T4, however, these were not significantly different ($p > 0.05$).

**Table 2.** Effect of coffee pulp increasing supplementation levels on milk yield, milk composition, live weight, body condition score, grass dry matter intake, and total dry matter intake.

| Treatment CoP kgDM | Milk Yield kg/d | Milk composition g/kg | | | Live Weight kg | BCS | GDMI kg DM/d | Proportion of CoP in Concentrate | Total DMI kg/d | Proportion of CoP in Total DMI |
|---|---|---|---|---|---|---|---|---|---|---|
| | | Fa | CP | TS | | | | | | |
| Treat. 1 (0) | 12.3 | 39.4 | 31.2 | 119.5 | 494 | 3.7 | 3.6 | 0.0 | 9.6 [a] | 0.0 |
| Treat. 2 (0.6) | 14.7 | 39.7 | 30.9 | 123.0 | 492 | 3.6 | 3.2 | 0.1 | 9.8 [a] | 0.062 |
| Treat. 3 (0.9) | 14.5 | 34.1 | 30.6 | 118.7 | 480 | 3.6 | 3.6 | 0.15 | 10.5 [b] | 0.085 |
| Treat. 4 (1.2) | 12.3 | 40.9 | 31.4 | 123.6 | 475 | 3.7 | 3.6 | 0.2 | 10.8 [b] | 0.11 |
| D.F. | 15 | 15 | 15 | 15 | 15 | 15 | 15 | - | 15 | - |
| SEM | 1.7 | 2.2 | 0.5 | 3.5 | 8.0 | 0.07 | 0.21 | - | 0.21 | - |
| P | 0.62 | 0.23 | 0.74 | 0.69 | 0.33 | 0.45 | 0.45 | - | 0.02 | - |

Means between rows with different letters were significantly different at the indicated *p*-value. CoP = coffee pulp, T = treatment, CP = crude protein in milk, TS = total solids in milk, BCS = body condition score, DMI = dry matter intake, GDMI = grass dry matter intake, SEM = standard error of the mean.

### 3.3. Sward Response Variables

The results for the African Star grass sward's performance are depicted for the experimental periods as sward variables were measured across all the experimental periods. However, the sward growth influenced the animal response to the treatments as shown in Figure 1, which illustrates the effect of the net forage accumulation on the average daily GDMI for each experimental period. It can be observed that NFA was above the 1500 kg DM/ha threshold, at which herbage intake is not affected, in all periods. The NFA in P1 was significantly higher ($p < 0.05$) than in the rest of the experimental periods. Results for NFA in Figure 1 suggest that NFA exerted more influence on GDMI than the CoP supplementation as indicated by the results for GDMI in Table 2, where no effect on GDMI ($p > 0.05$) was observed due to the inclusion of CoP at any of the supplementation levels tested. Figure 1 also shows that GDMI followed the same pattern of change as NFA across all experimental periods because the drop in NFA observed between P1 and P2 was followed by a reduction in GDMI by the animals. In the same way, the increase in NFA observed in P3 and P4 was accompanied by an increment in GDMI in the same experimental periods.

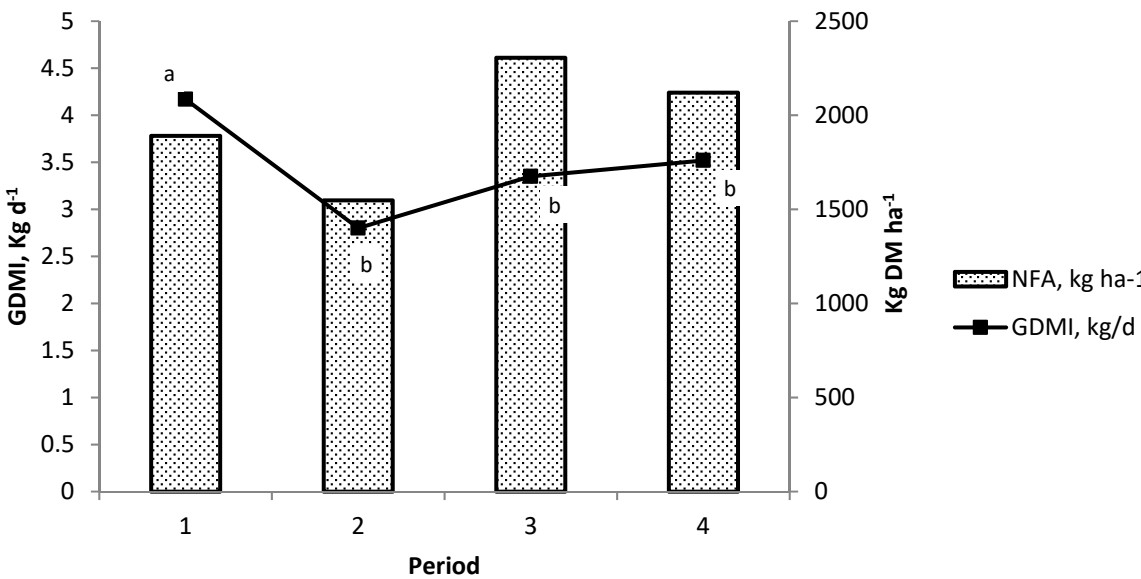

**Figure 1.** Influence of net forage accumulation (NFA, kg DM/ha) on average daily grass dry matter intake (GDMI, kg DM/head/d) of dual-purpose cows supplemented with increasing levels of coffee pulp across the four experimental periods. Different letters for GDMI were significantly different at $p < 0.05$.

Figure 2 shows that the average milk yield was also influenced by both the growth pattern of the grass as well as the crude protein content across the four experimental periods. This can be seen because the average milk yield per period numerically declined from P1 to P2 as a result of the NFA drop in the same periods. Similarly, when NFA and the crude protein content of the grass increased from P2 to P4, milk yield also increased. Although these changes in milk yield were not significant ($p > 0.05$), there was a numerical difference of more than two kilograms of milk produced between P2 and P4, which represents 23% more milk produced.

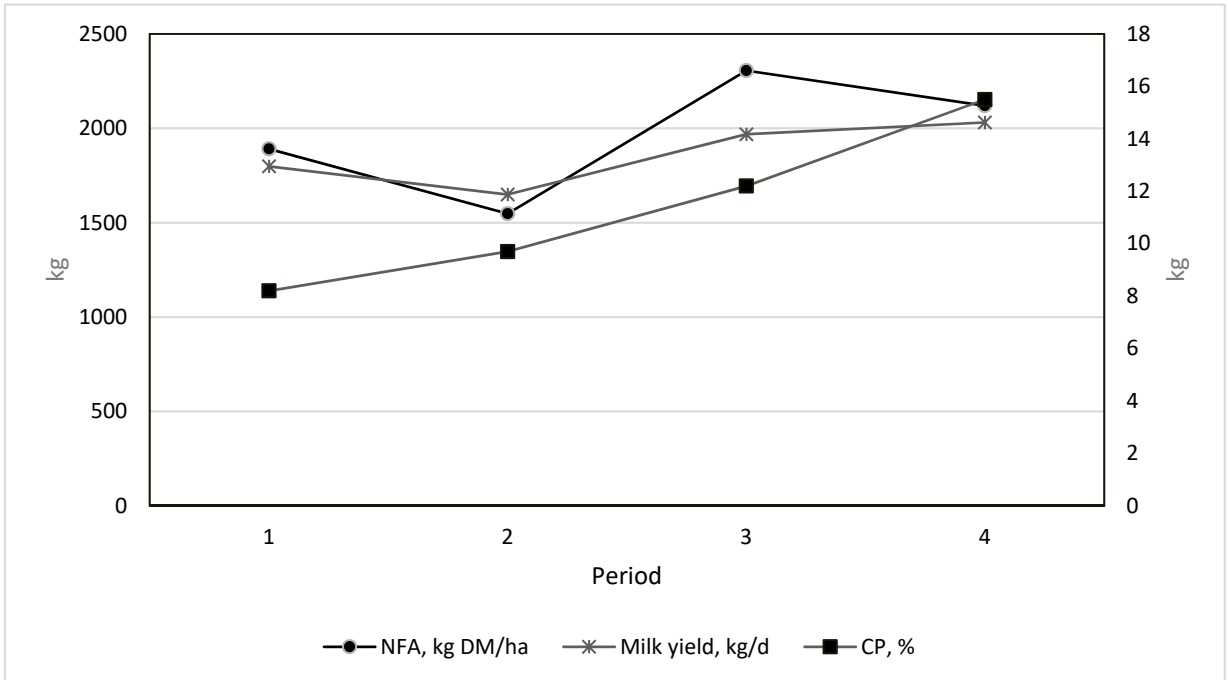

**Figure 2.** Effect of African Star grass net forage accumulation and crude protein content in the grass on average milk yield of dual-purpose cows supplemented with increasing levels of coffee pulp across all experimental periods.

## 4. Discussion

The chemical composition of the CoP was similar to that reported by Pandey and Soccol [32], particularly for tannins, polyphenols, and CP, but not for NDF, which was higher in our work (353 vs. 210 g/kg DM). This difference could be attributed to the processing system of the coffee cherry, which in the present work was conducted using the wet method followed by sun-drying for over two weeks. It has been demonstrated that the sun-drying method increases the NDF content in the CoP [33]. The caffeine content of our CoP was particularly low because an average content of 1% was reported by Asfew and Dekebo [34], with a maximum of up to 3% was reported by Ameca et al. [33]. This explains why no negative effects were observed on the behavior of the experimental animals due to the supplementation of CoP. On the other hand, the average CP content of the African Star grass sward (114 g/kg DM) was also similar to that reported by López-González et al. [35] for star grass swards harvested from April to July in Tejupilco, Mexico. However, the CP content was higher, up to 155 g/kg DM in P4, than in other tropical and subtropical grasses that normally present CP contents close to 90 g CP/kg DM [13], but still within the range reported for other Cynodon grass species [36]. The high CP content observed in P3 and P4 may be explained by the onset of the rains in June toward the end of the experiment, which resulted in an intense rate in the production of new shoots and leaves that were rich in protein (Table 1). The rainy season in the study area is from June to November [27], so after the end of the rains, the CP content of the grass in the study area returns to more normal values (e.g., <60 g CP/kg DM), as described by Yong-Angel [37] or the 82 g CP/kg DM observed in period 1 of the present study (Figure 2).

### 4.1. Effect of Coffee Pulp Supplementation on Milk Yield and Milk Composition of Dual-Purpose Cows

The inclusion of CoP showed no negative effect on milk yield or milk composition, suggesting that it may be included in up to 12% of dual-purpose cows' diets, as in T4, without compromising milk yield or milk composition. The highest CoP supplementation rate used in the present work was similar to the tolerable rates of supplementation recommended by Mazzafera [11] and Souza et al. [13] at 10.5% and 20% of the total DMI, respectively. Noriega-Salazar et al. [38] also agreed that CoP can be included in 10% to 20% of the total daily dry matter intake of a total mix ration for dairy cows without affecting milk yield. However, results for milk yield obtained in the present work suggest that low (0.6 kg DM/day) to medium CoP supplementation rates (0.9 kg DM/day) can produce better numerical milk yield results than the highest level of 1.2 kg DM/d. For example, low rates of CoP supplementation as in T2 resulted in 19.5% more milk produced (Table 2) than the highest rate used in T4. Numerically higher milk yield in T2 and T3 may be attributed to both the moderate content of tannins and phenols in the CoP [6], which probably increased the flow of the feed's protein to the duodenum of the animals, and to the extra supply of soluble carbohydrates provided by the CoP, which according to Klingel et al. [39], contains 45–89% of the total soluble carbohydrates of the coffee cherry. According to Nurfeta [12], low levels of tannins are linked to the feed's protein forming tannin–protein complexes, which act as a bypass protein that can be absorbed in the small intestine more efficiently. The increased flow of protein from the rumen to the lower gut has been associated with increased milk production in dairy cows [40,41], for example, Ibarra and Latrille [42] observed a significant increase ($p < 0.05$) in milk yield when Holstein cows were supplemented with fishmeal, a well-known source of bypass protein. Moreover, Chaves et al. [43] reported that feeding dietary phytochemicals (a combination of tannin mixture and capsicum) can significantly affect rumen fermentation characteristics in Holstein cows via partial manipulation of rumen microbiota and that these effects were reflected in improved milk production and efficiency. In contrast, the high supplementation level of CoP in T4 may have played the opposite role (e.g., increased tannin content in the

diet may have led to a larger formation of tannin–protein complexes, making protein from the diet less degradable in the rumen or undigestible in the rest of the digestive tract [44].

### 4.2. African Star Grass Herbage Intake and Concentrate Supplementation

The significant differences observed for total daily DMI (e.g., higher intake observed in T3 and T4 in comparison with T1 and T2) can be explained because cows in these treatments received 0.9 kg/DM and 1.2 kg DM/d of CoP, respectively, whereas cows in T1 did not receive CoP and cows in T2 received only 0.6 kg DM/d. However, it is important to stress that the effect of CoP supplementation does not explain the moderate to low GDMI (mean = 3.5 kg DM/d) observed across all treatments nor the forage availability in the experimental sward because the NFA was never below the threshold of 450–500 kg DM/ha at which the DMI of grazing cows becomes affected [45]. Therefore, it is likely that the low GDMI observed in this work was associated with two factors: (1) the limited number of hours the cows had access to the sward (<10 h/d); and (2) the concentrate supplementation rate used and its associative effects (e.g., substitution and reduced rumen pH). These hypotheses are supported by the fact that forage availability of the experimental sward was never below the 450–500 kg/ha threshold marked by Allison [45] at which grass DMI declines. According to Pérez-Ramírez et al. [46], Holstein cows at 577 kg of live weight with restricted access time to a ryegrass sward (9 h/d) showed a reduction in herbage intake and milk yield. These authors found that the reduction in milk production was due to a decrease in herbage intake of −1.8 kg DM/d, since the cows did not have enough time to reach their maximum intake. Concentrate supplementation rate can also affect herbage intake, for example, Pérez-Ramírez et al. [46] stated that high supplementation rates of 10 kg of concentrate DM/d/cow in Holstein cows of 578 kg of body weight had negative effects on herbage intake—less than 8.4 kg herbage DMI/d—because cows were not motivated to graze. In contrast, cows in Pérez-Ramírez et al.'s [46] experiment showed a stronger motivation for grazing (11.5 kg herbage DMI/d) when they received a low-supplement feeding regime of 5.0 kg of concentrate DM/d/cow). In the present work, the supplementation rate used possibly affected the motivation of cows to graze as in [46]; although our cows were less heavy than the cows used in Pérez-Ramírez's [46] experiment, they proportionally received a similar concentrate supplementation rate according to their body weight. As in the present work, Pedraza-Beltrán et al. [47] reported no negative effect of CoP supplementation on GDMI, milk yield, and milk composition ($p > 0.05$) of dual-purpose cows grazing on a Bahía grass (*Paspalum notatum*) sward located in the same region. Moreover, according to Pedraza-Beltrán et al. [47], CoP can replace some of the purchased ingredients of dairy concentrates. The supplementation rate of the experimental concentrate used could also have limited grass intake, but to a lesser extent by a reduction in rumen pH since important drops in low-quality forage intake by dairy cattle in Tanzania have been observed following supplementation with high amounts of non-fibrous carbohydrates such as starch [48]. It is well established that low rumen pH reduces the capacity of the rumen microbes to degrade the fiber in forage [49]. Decreasing the fiber fermentation as a consequence of low rumen pH and the increased residence time of feed in the rumen results in animal satiety and temporary detention of voluntary intake. However, more research is needed in the case of tropical grasses and tropical cattle because most of the work published on this subject has been done for temperate grasses, and there is little evidence in the literature on this particular topic.

Our results suggest that low supplementation rates of CoP worked better than high supplementation rates, and that CoP could be an alternative way to reduce the costs associated with feeding dual-purpose cattle because CoP has no cost other than transportation to the farms. Reducing the feeding costs of cattle can improve the income of smallholder farmers and contribute to improving their livelihoods. The use of CoP as cattle feed can also contribute to mitigating the environmental damage produced by inadequate disposal of CoP during coffee production in tropical and subtropical regions around the world. Finally, we suggest that farmers allocate more time for cattle to graze on the African Star

grass swards, possibly up to 24 h, to maximize grass intake by cows, however, the insecurity remains a problem that needs to be solved before implementing such measures.

## 5. Conclusions

Within the limits of the present study, it can be concluded that increasing the supplementation levels of coffee pulp did not have negative effects on milk yield, milk composition, and grass intake. However, numerical increments in milk yield were observed when CoP was supplemented at doses of 0.6 and 0.9 kg DM/d, suggesting that low doses of CoP could increase milk yield, possibly due to the moderate intake of tannins and their potential role in increasing the flow of the feed's protein to the duodenum of the animals, which was associated with the extra supply of soluble carbohydrates by the CoP. However, additional research using more animals is necessary to confirm these results. The lack of cost of CoP is an additional benefit because similar works suggest that CoP can replace some purchased ingredients of dairy concentrates. The results of our work also suggest that the use of CoP to supplement dual-purpose cattle can be an alternative method for the safe disposal and sustainable management of CoP in the tropics and may help in reducing production costs associated with cattle feeding.

**Author Contributions:** Conceptualization, O.A.C.-O.; Methodology, O.A.C.-O., F.A.-N., and J.G.E.-F.; Investigation, P.E.P.-B., G.Y.-Á., A.D.S.-M., and A.-A.R.-A.; Resources O.A.C.-O.; Data curation, P.E.P.-B. and M.F.V.-C.; Writing—original draft preparation, O.A.C.-O.; Writing—review and editing, O.A.C.-O. and M.G.-R.; Project administration, O.A.C.-O. and J.G.E.-F.; Funding acquisition, O.A.C.-O. All authors have read and agreed to the published version of the manuscript.

**Funding:** This research was funded by the ICAMEX-Estado de México under grant number 15-2005-0724 and the Universidad Autónoma del Estado de México under grant number 1873/2009C.

**Institutional Review Board Statement:** All experimental procedures were approved by the Committee on Bioethics and Animal Welfare of the Universidad Autónoma del Estado de México (UAEMex), and the Mexican norm for work with laboratory animals NOM-062-ZOO-1999. No suffering was inflicted on the experimental animals.

**Informed Consent Statement:** Informed consent was obtained from all subjects involved in the study.

**Data Availability Statement:** The data presented in this study are available on request from the corresponding author.

**Acknowledgments:** We acknowledge the help of cattle farmers from the municipality of Tejupilco and the coffee producers of the municipality of Amatepec, Mexico whose help was essential for the conclusion of the present work.

**Conflicts of Interest:** The authors declare no conflicts of interest.

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
