# Peer review of "Effect of Increasing Supplementation Levels of Coffee Pulp on Milk Yield and Food Intake in Dual-Purpose Cows: An Alternative Feed Byproduct for Smallholder Dairy Systems of Tropical Climate Regions"

_agriculture, doi:10.3390/agriculture11050416_

Round 1
Reviewer 1 Report
Dear Authors,
thank you for taking into consideration my previous comments, I appreciate it. Speaking of the "non significant" results of the study (Milk yeld, milk composition etc.), they shouldn't affect the quality of the work, however a small number of individuals sometimes may not ensure sufficient statistical power. According to the Journal of Dairy Science Instructions to Authors (2021) a “non significant” relationship should not be interpreted to suggest the absence of a relationship. However an inadequate number of experimental units or insufficient control of variation, limits the power to detect relationships. The latin square system with 4 cows could be affected by the individual response of just one cow. However, according to the Authors and bibliography, I totally agree on the importance of a moderate amount of tannins in dairy cattle diet. Generally speaking, a low level integrations of tannins from coffee pulp could be a very important results for cows and the echosystem in terms of re-using resources with zero economic and environmental impact.
Author Response
Response to reviewer #1, second round
thank you for taking into consideration my previous comments, I appreciate it. Speaking of the "non significant" results of the study (Milk yeld, milk composition etc.), they shouldn't affect the quality of the work, however a small number of individuals sometimes may not ensure sufficient statistical power. According to the Journal of Dairy Science Instructions to Authors (2021) a “non significant” relationship should not be interpreted to suggest the absence of a relationship. However an inadequate number of experimental units or insufficient control of variation, limits the power to detect relationships. The latin square system with 4 cows could be affected by the individual response of just one cow. However, according to the Authors and bibliography, I totally agree on the importance of a moderate amount of tannins in dairy cattle diet. Generally speaking, a low level integrations of tannins from coffee pulp could be a very important results for cows and the echosystem in terms of re-using resources with zero economic and environmental impact.
Authors’ Response: Dear reviewer we are pleased to hear that you are happy with the way we addressed your comments. You are right, it is important to report large numerical differences even if they are not statistically significant. Unfortunately, sometimes it is not possible to get funding to increase the number of animals, particularly with cows, that without any doubt will allow finding significant differences. We agree with you that the response of one single animal can affect the final result by increasing the variation in a way that no significant differences can be observed. We addressed this point in the conclusion saying that it is necessary to conduct more experiments using more animals.
Last but not least, it is worth mentioning that several experiments have been conducted and published in relevant journals using a single 4 x 4 Latin Square Experimental design, here are some examples:
Hammond, K.J. & Humphries, D.J. & Crompton, L.A. & Reynolds, Christopher. (2015). Methane emissions from cattle: Estimates from short-term measurements using a GreenFeed system compared with measurements obtained using respiration chambers or sulphur hexafluoride tracer. Animal Feed Science and Technology. 203. 10.1016/j.anifeedsci.2015.02.008
The Effects of Three Herbs as Feed Supplements on Blood Metabolites, Hormones, Antioxidant Activity, IgG Concentration, and Ruminal Fermentation in Holstein Steers . K. Hosoda, K. Kuramoto, B. Eruden, T. Nishida, S. Shioya. Asian-Australasian Journal of Animal Sciences 2006;19(1): 35-41. DOI: https://doi.org/10.5713/ajas.2006.35
Suksombat W, Nanon A, Meeprom C, Lounglawan P. Feed degradability, rumen fermentation and blood metabolites in response to essential oil addition to fistulated non-lactating dairy cow diets. Anim Sci J. 2017;88(9):1346-1351. doi:10.1111/asj.12778

Reviewer 2 Report
The research topic is an advantage of the publication.
The manuscript presented for evaluation covers a issues related to the potential use of coffee pulp a coffee processing by-product as an alternative feed in cattle feeding. It is a response to the great interest in scientific information among smallholder-crop-livestock farmers. This information would allow the use of larger amounts of these products in animal feeding, maybe for partially replacing traditional components such as cereal grains. The adopted research assumptions are important, valuable and interesting.
The manuscript is written carefully and introduction and discussion based on relevant and up-to-date scientific literature, but the interpretation of the obtained results is questionable
The technical form of the manuscript submitted for evaluation is satisfactory, but some chapters require improving the assumptions and scientific interpretation of the obtained observation results.
Correctly presented chapters:
- The introduction,
- The methodology (except for the description of the composition of daily feed rations and of the experimental concentrates (see notes in text);
- The discussion of the differences in the analyzed indicators.
Fragments of insufficient quality:
- The chapters: Abstract, Results (including Tables) and Conclusions do not meet the requirements;do not live up to expectations
- Abstract- logical inconsistencies in the text were indicated;
- Results- tables and graphs should be rethought and redrafted, in the presented form are incomprehensible (see notes in table 1);
- Conclusions - try to answer the question: cows without or with additive achieved the same values of the analyzed indicators, on what basis is the additive recommended. This aspect of the research is fundamental.
Some comments were attached to the text as examples of sources of doubts arising from the reader of the study

Author Response
Response to reviewer #2, first round.
Reviewer # 2: The manuscript presented for evaluation covers a issues related to the potential use of coffee pulp a coffee processing by-product as an alternative feed in cattle feeding. It is a response to the great interest in scientific information among smallholder-crop-livestock farmers. This information would allow the use of larger amounts of these products in animal feeding, maybe for partially replacing traditional components such as cereal grains. The adopted research assumptions are important, valuable and interesting.
The manuscript is written carefully and introduction and discussion based on relevant and up-to-date scientific literature, but the interpretation of the obtained results is questionable.
The technical form of the manuscript submitted for evaluation is satisfactory, but some chapters require improving the assumptions and scientific interpretation of the obtained observation results.
Authors response: Dear Reviewer #2
Thank you very much for using some of your valuable time for reading and commenting on our manuscript. We believe that all your comments and suggestions helped us to elaborate a better manuscript. We think we addressed all the points raised by you, and in the sections below we responded to each of your questions. We also made the necessary changes directly to the manuscript and responded to all your question in the pdf file uploaded together with your comments. You will be able to see the changes and amendments because they are marked by the Word tool to track changes. We hope we were able to meet your expectations and queries; however, we will be happy to answer any new questions that you may have.
Reviewer #2: The methodology (except for the description of the composition of daily feed rations and of the experimental concentrates (see notes in text).
Response: We already specified that the coffee pulp was supplemented on top of the concentrate, so to know the proportion of CoP in the concentrate the reader just need to do a simple calculation, e.g. =(0.6/6.0)*100, or =(0.9/6.0)*100. We mentioned in the manuscript that cows ate all the concentrate offered, so 6.0 kg is a fixed value. The proportion of the CoP in the daily ration (average total dry matter intake) depended on the herbage intake because this differed between treatments.
However, to address the kind suggestion of the reviewer we inserted a new column in Table 2 that shows the percentage of CoP in the whole diet, the column is named: Proportion of CoP in Total DMI.
Reviewer #2: The discussion of the differences in the analyzed indicators.
Response: We improved the discussion for the variables mentioned by the reviewer and make more clear that we only found significant differences for average total daily dry matter intake in treatments T3 and T4 in comparison with treatments T1 and T2.
Reviewer #2: Fragments of insufficient quality: The chapters: Abstract, Results (including Tables) and Conclusions do not meet the requirements; do not live up to expectations
- Abstract- logical inconsistencies in the text were indicated;
Response: We addressed all the inconsistencies in the abstract, please see lines 10 to 16 of the abstract.
Reviewer #2: Results- tables and graphs should be rethought and redrafted, in the presented form are incomprehensible (see notes in table 1);
Response: we redrafted the tables as indicated by the reviewer, all figures are expressed now in g/kg DM otherwise units are clearly stated. We also inserted a new column in Table 2 that shows the percentage of CoP in the whole diet.
Reviewer #2: Conclusions - try to answer the question: cows without or with additive achieved the same values of the analyzed indicators, on what basis is the additive recommended. This aspect of the research is fundamental.
Response: we rewrote the conclusion to address the comments of the reviewer, it now reads as follows:
Within the limits of the present study, it can be concluded that increasing supplementation levels of coffee pulp did not have negative effects on milk yield, milk composition, and grass intake. However, numerical increments in milk yield were observed when CoP was supplemented at doses of 0.6 and 0.9 kg DM/d, suggesting that low doses of CoP could increase milk yield possibly due to the moderate intake of tannins on increasing the flow of feed’s protein to the duodenum of the animals, and the extra supply of soluble carbohydrates provided by the CoP. However, additional research using more animals is necessary to confirm these results. The lack of cost of CoP is an additional benefit because similar works suggest that CoP can replace some purchased ingredients of dairy concentrates. The results of our work also suggest that the use of CoP to supplementing dual-purpose cattle can be an alternative way for the safe disposal and sustainable management of CoP in the tropics and may help in reducing production costs associated with cattle feeding.
Reviewer #2: Some comments were attached to the text as examples of sources of doubts arising from the reader of the study
Response: We responded to all comments directly on the pdf file that is submitted with the reviewed version of our manuscript.

Round 2
Reviewer 2 Report
In my opinion, the authors did not correct the manuscript submitted for evaluation insufficiently carefully.
For example, these important passages have not been sufficiently re-written:
in relation to the title
1 - if: no unequivocal information about the study of the impact (effects...) of .. (factor X) on .. (tested indicator)
4 - I suggest: tropical climate regions,
in relation to the chapters Abstract and Materials and Methods
33 – there is no definition of P-value accepted as a statistically significant,
an important determinant of statistical evaluation is the distinction between statistically significant differences and numerical differences (what does it mean?).
25 – a fixed rate – delete,
34 – 35 - I propose to re-rank according to generally accepted principles of keyword selection,
268 - I propose to start with the characterization of the research object, coffee pulp,
in relation to the Results
tables 1 and 2
- data presented in the table are not sufficiently explained, e.g. P1 ..P4,
- are the chemical components of the feed also e.g. tannins etc.
.,
- please complete with specific values of the share of coffee pulp in DM of concentrated mix or daily ration (it is easy to calculate..),
Figures 1 and 2
- the presented data was not clearly described,
- figure 1 remained absolutely unclear, it does not explain the relationship between the amount of coffee pulp added and the roughage intake- i.e.
Please note that each table and each figure should constitute a separate information unit, understandable to an outside reader.
Round 3
Reviewer #2: In my opinion, the authors did not correct the manuscript submitted for evaluation insufficiently carefully.
Response: Dear reviewer please accept our apologies for not addressing sufficiently your comments. We paid more attention to responding to each one of your queries as we acknowledge they help to improve our manuscript. We also thank you for the extra time you allocated to read our manuscript.
Reviewer #2: For example, these important passages have not been sufficiently re-written:
In relation to the title
1 - if: no unequivocal information about the study of the impact (effects...) of .. (factor X) on .. (tested indicator)
4 - I suggest: tropical climate regions,
Response: We have rewritten the title taking into account: the effect of supplementing coffee pulp on milk yield and food intake as testing indicators. We also inserted the words “tropical climate regions” in the title.
Reviewer #2: in relation to the chapters Abstract and Materials and Methods
33 – there is no definition of P-value accepted as a statistically significant, an important determinant of statistical evaluation is the distinction between statistically significant differences and numerical differences (what does it mean?).
Response: we now provided the significant level at which the statistical difference was observed. We consider it convenient to mention the numerical differences observed for milk yield because these represent 19.5% and 17.8% more milk, produced in treatment 2 and treatment 3, respectively in comparison with the control treatment. We also mentioned that if we had increased the number of animals we will probably find a significant difference.
Reviewer #2: 25 – a fixed rate – delete,
Response: We deleted fixed rate.
Reviewer #2: 34 – 35 - I propose to re-rank according to generally accepted principles of keyword selection, 268 - I propose to start with the characterization of the research object, coffee pulp, in relation to the Results tables 1 and 2
Response: We re-ranked the keywords according to MDPI-Agriculture Journal, Springer Journals, Elsevier author services, which suggests the following criteria to choose effective keywords:
- Represent the content of your manuscript
- Be specific to your field or sub-field
- Avoiding the duplication of words already in the article’s title is strongly recommended by journals. It is preferable to choose keywords that complement the main topic of your research, including related words and/or methodology-specific terms.
- According to MDPI Agriculture, the keywords should be three to ten pertinent keywords that need to be added after the abstract. We recommend that the keywords are specific to the article, yet reasonably common within the subject discipline.
Based on the above recommendation we rewrote and re-ranked our keywords starting by “Coffee pulp” as follows:
Keywords: Coffee pulp, Cynodon plectostachius; milk composition, tannins; polyphenols, by-pass protein; sustainable cattle production; local feed resources, coffee waste products, substitution rate.
Reviewer #2: - data presented in the table are not sufficiently explained, e.g. P1 ..P4, - are the chemical components of the feed also e.g. tannins etc.
Response: We replaced the letter “P” with the word “Period” to make it clearer that we are referring to the experimental period, and added a heading with the words “Experimental Periods” too. We rephrased the title of the table to separate the description of chemical components from plants secondary metabolites of coffee pulp.
Reviewer #2: - please complete with specific values of the share of coffee pulp in DM of concentrated mix or daily ration (it is easy to calculate..),
Response: We inserted a new column to indicate the proportion of coffee pulp in the experimental concentrate, and in the previous round we inserted a column to indicate the proportion of coffee pulp in total DMI. Thank you very much for your comment.
Reviewer #2: Figures 1 and 2. - the presented data was not clearly described, - figure 1 remained absolutely unclear, it does not explain the relationship between the amount of coffee pulp added and the roughage intake- i.e.
Response: The idea behind these figures is to demonstrate that the herbage intake, which is the main source of foodstuff for grazing cattle, was not affected by the supplementation with coffee pulp. This is why you can not see the effect of CoP in these figures, in addition, the performance of the swards was evaluated in periods. This is a relevant point for our work because as you can see the grass intakes were moderate to low, so readers may want to attribute this to the coffee pulp. We tried to provide the elements for readers, so they can see/evaluate that coffee pulp, within the limits of the present study, did not affect herbage intake. And, that other factors affected herbage intake, like the short time allocated for grazing and possible substitution effect of concentrate. We checked this section again a tried to clarify this point. Hopefully, we achieve this purpose this time.
Reviewer #2: Please note that each table and each figure should constitute a separate information unit, understandable to an outside reader.
Response: We corrected the titles and figures as suggested by the reviewer.
